# De novo emergence of a remdesivir resistance mutation during treatment of persistent SARS-CoV-2 infection in an immunocompromised patient: a case report

Shiv Gandhi [1,12✉], Jonathan Klein [2,12], Alexander J. Robertson [3,12], Mario A. Peña-Hernández[2,12],
Michelle J. Lin [4], Pavitra Roychoudhury[4], Peiwen Lu [2], John Fournier[1], David Ferguson[5],
Shah A. K. Mohamed Bakhash[4], M. Catherine Muenker [3], Ariktha Srivathsan[3], Elsio A. Wunder Jr [3],
Nicholas Kerantzas[6], Wenshuai Wang[7], Brett Lindenbach[8], Anna Pyle[7,9,10], Craig B. Wilen [2,6],
Onyema Ogbuagu[1], Alexander L. Greninger[4,11], Akiko Iwasaki [2,3,7,10], Wade L. Schulz [5,6] &
Albert I. Ko [1,3✉]

SARS-CoV-2 remdesivir resistance mutations have been generated in vitro but have not been reported in patients receiving treatment with the antiviral agent. We present a case of an immunocompromised patient with acquired B-cell deficiency who developed an indolent, protracted course of SARS-CoV-2 infection. Remdesivir therapy alleviated symptoms and produced a transient virologic response, but her course was complicated by recrudescence of high-grade viral shedding. Whole genome sequencing identified a mutation, E802D, in the nsp12 RNA-dependent RNA polymerase, which was not present in pre-treatment specimens. In vitro experiments demonstrated that the mutation conferred a ~6-fold increase in remdesivir $IC_{50}$ but resulted in a fitness cost in the absence of remdesivir. Sustained clinical and virologic response was achieved after treatment with casirivimab-imdevimab. Although the fitness cost observed in vitro may limit the risk posed by E802D, this case illustrates the importance of monitoring for remdesivir resistance and the potential benefit of combinatorial therapies in immunocompromised patients with SARS-CoV-2 infection.

[1] Section of Infectious Diseases, Department of Medicine, Yale University School of Medicine, New Haven, CT, USA. [2] Department of Immunobiology, Yale University School of Medicine, New Haven, CT, USA. [3] Department of Epidemiology of Microbial Diseases, Yale School of Public Health, New Haven, CT, USA. [4] Department of Laboratory Medicine, University of Washington School of Medicine, Seattle, WA, USA. [5] Center for Outcomes Research and Evaluation, Yale New Haven Hospital, New Haven, CT, USA. [6] Department of Laboratory Medicine, Yale School of Medicine, New Haven, CT, USA. [7] Department of Molecular, Cellular and Developmental Biology, Yale University, New Haven, CT, USA. [8] Department of Microbial Pathogenesis, Yale School of Medicine, New Haven, CT, USA. [9] Department of Chemistry, Yale University, New Haven, CT, USA. [10] Howard Hughes Medical Institute, Chevy Chase, MD, USA. [11] Vaccine and Infectious Disease Division, Fred Hutchinson Cancer Research Center, Seattle, WA, USA. [12] These authors contributed equally: Shiv Gandhi, Jonathan Klein, Alexander Robertson, Mario A. Peña-Hernández. ✉email: shiv.gandhi@yale.edu; albert.ko@yale.edu

Remdesivir (RDV), an adenosine nucleoside analog that interferes with the SARS-CoV-2 RNA polymerase, nsp12, when converted to its active form[1], has been widely used in the treatment of patients hospitalized with COVID-19. In vitro selection experiments have generated mutations at residue 802 of nps12, which confer resistance to RDV[2]. These mutations have been detected in genome sequences deposited in GISAID[2], but their emergence has not been demonstrated in patients receiving RDV therapy. Selection for RDV resistance is a specific concern for immunocompromised patients who may receive multiple courses of the antiviral agent for persistent SARS-CoV-2 infection[3–5]. Here, we report a case of an immunocompromised patient with an indolent, protracted course of SARS-CoV-2 infection from whom a RDV resistance mutation, E802D, was identified during recrudescence of viral shedding following treatment with the antiviral agent.

## Results

**Persistent COVID-19 infection in a patient with acquired B-cell deficiency.** A 70-year-old woman with Stage IV Non-Hodgkin's lymphoma (NHL) had completed a course of rituximab and bendamustine in March 2019, which was complicated by lymphocytopenia and hypogammaglobinemia. The NHL was in remission when she developed a SARS-CoV-2 infection in May 2020, which presented as the acute onset of fever, anosmia, cough and rhinorrhea (Fig. 1a) and was confirmed by reverse transcriptase-polymerase chain reaction (RT-PCR) on day 0 (Fig. 1b).

The patient was hospitalized twice in the following two months for persistent fever and new onset neutropenia and anemia (Fig. 1c), during which she had low SARS-CoV-2 RT-PCR cycle threshold (Ct) values from nasopharyngeal specimens (Ct 20.5 and 17.1 on days 17 and 36, respectively, Fig. 1b), undetectable serum anti-SARS-CoV-2 IgG antibody (Abbott), and elevated serum inflammatory markers (Fig. 1c). Chest computed tomography (CT) revealed bilateral ground glass opacities (day 38, Fig. 1d), but she did not experience dyspnea, hypoxemia or recurrence of respiratory symptoms during her hospitalizations and subsequent course of illness. The workup for other sources of fever, which included a bone marrow biopsy, was unrevealing. CD19+ B-cells were not identified in the bone marrow biopsy or by flow-cytometry of peripheral blood mononuclear cells (PBMC). The patient was initiated on treatment with filgrastim 1–2 times per week and monthly intravenous immunoglobulin (IVIG) to manage her neutropenia and hypogammaglobulinemia, respectively, and both were continued after hospital discharge on day 51.

The patient continued to have daily fevers, except for a 30-day period of defervescence (days 103–132), refractory neutropenia, anemia, and positive RT-PCR test results with Ct values <25. Her chest CT on day 140 was notable for increased opacities compared with prior examinations (Fig. 1d).

**Clinical and virological response to remdesivir and mAb therapy.** The patient was hospitalized at our institution and received ten-day course of RDV (day 148–157), which was extended from the standard five-day regimen, because of concerns for clinical and virologic relapse in immunocompromised hosts with persistent infection. The patient had a clinical response to therapy, characterized by resolution of her fever (day 149, Fig. 1a), normalization of CRP (day 156, Fig. 1c) and improvement in the opacities on chest CT (day 162, Fig. 1d). She also had an initial virologic response to therapy with an increase in RT-PCR Ct values, decreasing numbers of plaque forming units (PFU) in viral cultures and lower proportions of SARS-CoV-2

subgenomic RNA (sgRNA), as measured in amplicon sequencing data, in her respiratory tract specimens by day 152 (Fig. 2a).

However, a recrudescence of viral shedding occurred during and after RDV therapy, as evidenced by decreasing RT-PCR Ct values which reached 18 on day 160 and increasing PFU and sgRNA in respiratory tract specimens on day 156 (Fig. 2a). The patient received an 8 g infusion of casirivimab-imdevimab[6,7] on day 163 after expanded access for the use of the monoclonal antibody (mAb) therapy was approved. A rapid and sustained virologic response was observed with undetectable cultured virus, low sgRNA and negative NP RNA in specimens by days 164, 166 and 217, respectively. The patient's anosmia resolved roughly 17 days after administration of casirivimab-imdevimab (day 180. Fig. 1a). During the following five-month period (days 160–292), the patient did not have recurrence of COVID-19 related symptoms and had a chest CT which showed minimal opacities (day 170, Fig. 1d), and had high anti-SARS-CoV-2 S1 protein IgG titers (Fig. 2a). Peripheral blood neutrophil counts, hemoglobin levels and serum inflammatory markers normalized during convalescence (Fig. 1c).

**Emergence of the nsp12 E802D mutation during remdesivir therapy.** To identify SARS-CoV-2 mutations that arose during the course of illness, we longitudinally sampled multiple sources of the patient's tissues and secretions (Supplementary Table 1) and performed WGS on an Illumina NextSeq platform using the Swift SARS-CoV-2 multiplex amplicon sequencing panel[8]. Viral N1 or N2 RNA was detected in 19 of 21 nasopharyngeal, 10 of 10 saliva, 10 of 10 stool and 0 of 10 whole blood specimens. Among the 39 samples with detectable N1 or N2 RNA, we assembled 27 whole SARS-CoV-2 genomes from nasopharyngeal (12), saliva (9) and stool (6) specimens. One of the nasopharyngeal specimens was obtained from the initial phase of illness (day 36, Fig. 2a). Phylogenetic analysis found that sequenced genomes belonged to a single lineage (Pango B1) within Nextstrain clade 20 C (Supplementary Fig. 1), indicating that viral genomes identified during the course of illness were derived from intra-host diversification following infection with a single strain.

Analysis of assembled viral genomes identified a mutation, E802D, in nsp12, whose detection in patient specimens was temporally associated with RDV therapy (Fig. 2b). E802D was not identified at an allele frequency above 1% in either the specimen obtained during the initial phase of illness (day 36) or in specimens collected during the first 5 days (days 148–152) of RDV therapy. The mutation was first detected 7 days after initiation of RDV therapy (day 155) and accounted for 23% and 96% of the allele frequency in nasopharyngeal and saliva specimens, respectively, by day 160. In a PAML analysis[9] of the Orf1a/1ab reading frame, this position was identified as having the highest probability (BEB Prob(w > 1), 0.984; mean omega, 9.90) for positive selection after RDV administration.

The E802 residue of nsp12 resides in the palm sub-domain which contains a portion of the residues that comprise the active site of the SARS-CoV-2 RNA polymerase[1,2,10]. The E802 residue participates in an electrostatic network with D804 and K807 which stabilizes the loop involved with binding to the nascent RNA (Fig. 2c, Supplementary Fig. 2a). E802D has been identified in an in vitro RDV resistance selection experiment and was found to confer a ~2.5-fold increase in $IC_{50}$ to the drug[2]. Together, the temporality of E802D emergence in the patient, its location in nsp12 and the in vitro identification of the same mutation with a RDV resistance phenotype support the plausibility that that RDV treatment of the patient selected for variants with the E802D mutation, which in turn contributed to the observed rebound viral shedding.

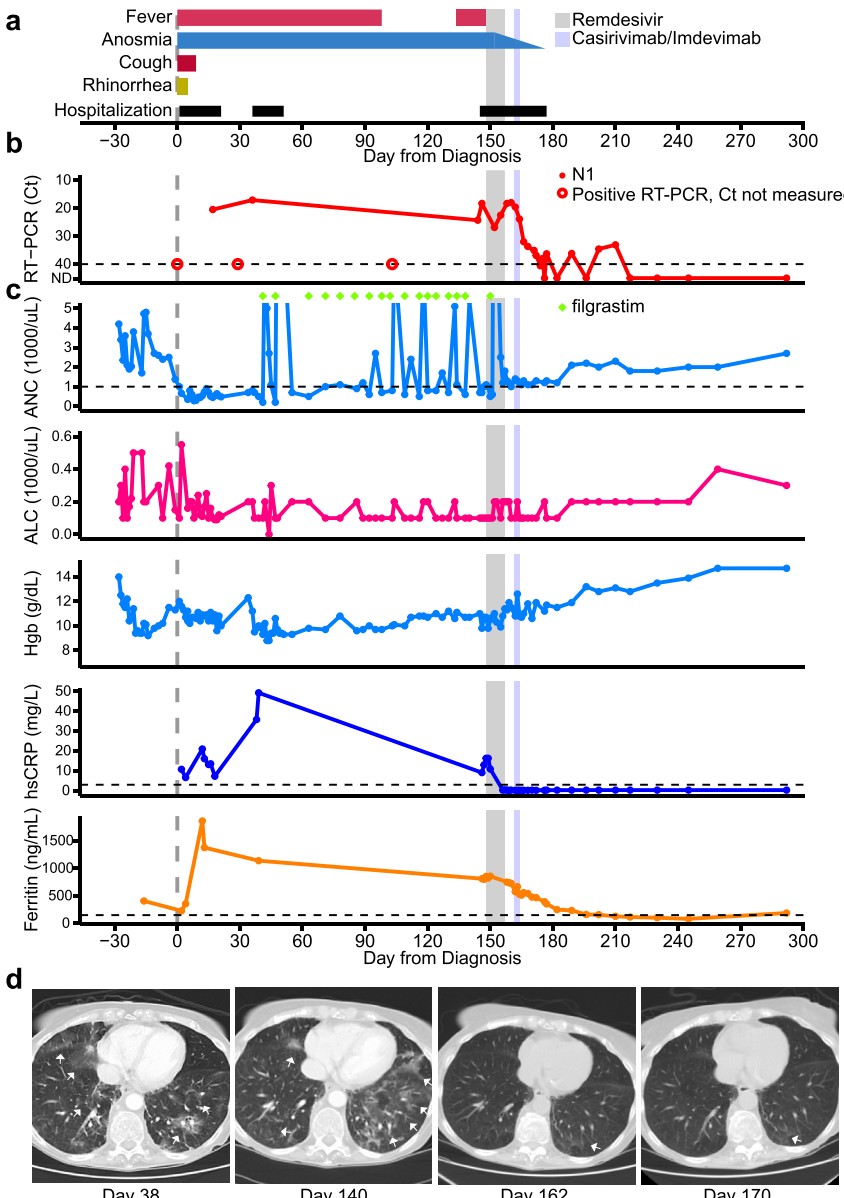

**Fig. 1 Clinical course of the immunocompromised patient with persistent SARS-CoV-2 infection.** Timeline of (**a**) patient symptoms and hospitalizations, (**b**) SARS-CoV-2 N1 RT-PCR Ct values and (**c**) clinical laboratory parameters from time of laboratory diagnosis of SARS-CoV-2 infection (day 0) to end of the follow-up period (day 292) and (**d**) computed tomography (CT) scans of chest at indicated days after time of initial diagnosis. The timing of remdesivir and casirivimab-imdevimab are shown as gray and light blue shading, respectively. RT-PCR results that were positive but performed on assays that did not generate a Ct value are denoted by the open circle in panel (**b**). The timing of filgrastim treatments are denoted by green diamonds in panel (**c**). lsCRP values were converted to hsCRP values using a correction factor of 9.2. Ground-glass opacities marked by white arrows in panel (**d**). Abbreviations: Real-time polymerase chain reaction (RT-PCR); Cycle threshold (Ct); Absolute neutrophil count (ANC); Absolute lymphocyte count (ALC); Hemoglobin (Hgb); high-sensitivity C-reactive protein (hsCRP).

**In vitro validation of nsp12 E802D as a remdesivir resistance mutation**. To validate the resistance phenotype of E802D, we engineered this mutation and an E802A mutation, which had been also shown to confer RDV resistance[2], into an infectious molecular clone of SARS-CoV-2/WA01 (icSARS-CoV-2-mNG), which expresses the mNeon Green reporter and is ORF7a depleted[11]. Replication kinetics of the E802D and E802A mutants in Vero-E6 cells revealed decreased viral replication relative to parental icSARS-CoV-2-mNG (Fig. 2d), suggesting that these mutations may impart a fitness cost. In the presence of high concentrations of RDV (5 μM), the E802D mutant replicated to higher titers than the parental virus (Supplementary Fig. 2b), confirming that this mutation confers resistance to RDV. RDV

dose-response curves demonstrated that E802D and E802A mutants had significant increases in $IC_{50}$ values (4.2 μM and 2.7 μM, respectively), relative to parental icSARS-CoV-2-mNG (0.7 μM) (Fig. 2e). Assessment of RDV cytotoxicity demonstrated limited impact on cell death rates below 10 μM, indicating that the RDV resistance phenotype confered by mutations at E802 may occur at physiologically relevant concentrations (Supplementary Fig. 2c).

These findings suggest that RDV resistance mediated by substitutions at residue 802 in nsp12 may lead to therapeutic failure in the setting of an immunocompromised host. We speculate that mutations at 802 distort the active site in a way that either enables the enzyme to exclude RDV or alleviates the steric

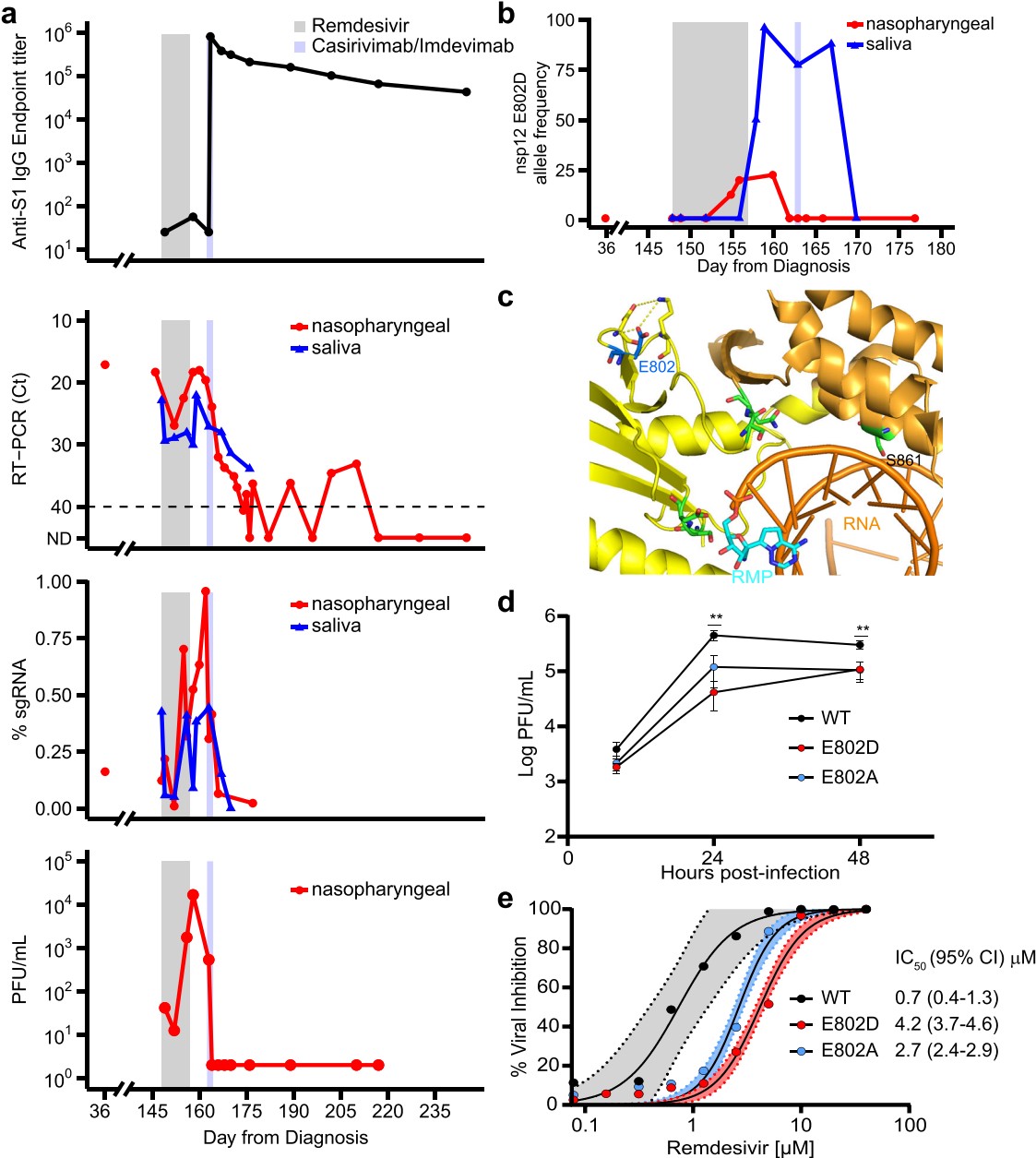

**Fig. 2 De novo emergence of remdesivir resistance mutation during and following treatment with the antiviral agent. a** Anti-SARS-CoV-2 Spike Protein IgG ELISA endpoint titers (1st panel), RT-PCR Ct values (2nd panel), subgenomic RNA (sgRNA, 3rd panel), and plaque forming units (PFU) on viral culture (4th panel) during the course of illness which include the period when remdesivir (gray shading) and casirivimab/imdevimab (light blue shading) were administered. **b** Allele frequencies of E802D in nsp12 as ascertained by whole genome sequencing. **c** Crystal structure (PDB:7BV2) depicting nsp12 E802 (dark blue), its putative hydrogen bonds (yellow dashed lines) within the palm domain (yellow), and the residues (green) that interact with the replicating RNA molecule (orange). Remdesivir monophosphate (RMP) is depicted in light blue. **d** Viral growth curves from icSARS-CoV-2 mNG WT, E802D (patient), and E802A (control) nsp12 mutants on a ORF7a depleted backbone. Results are depicted as mean and standard deviation (error bars) of data from biological replicates ($n = 3$). Significance between WT and E802D mutants was assessed by unpaired, two-sample $t$ tests, **$p < 0.01$ ($p = 0.002$ at 24 h, $p = 0.006$ at 48 h). **e** SARS-CoV-2 inhibition by RDV assessed at 48 h post-infection (0.01 multiplicity of infection) as determined by quantitative image analysis of percentage of cells expressing mNG. Color-coded curves represent a non-linear least squares fit of biological replicates ($n = 3$) and shading represents 95% confidence intervals of the fit. Figure is representative of four biological experiments.

clash mediated by S861, thereby allowing the enzyme to escape RDV-mediated chain termination[1,10]. Further biochemical and structural characterization will be needed to delineate the mechanism by which substitutions at residue E802 confer resistance to RDV.

The therapeutic impact of RDV resistance mutations at E802 may be attenuated by a fitness cost imparted by these mutations, which we observed in vitro. Although RDV therapy has been widely administered to patients during the pandemic, E802D and all substitutions at residue 802 have been found in 131 and 297, respectively, of the 4.8 M genome sequences obtained from patient isolates in the GISAID database (www.gisaid.org; accessed 11/07/21). Consistent with the observation of a fitness cost, the allele frequency of E802D in nasopharyngeal specimens from the patient decreased from 22.6% to <1% between the period (day 160–162) after completion of RDV therapy and prior to

casirivimab-imdevimab administration (Fig. 2b). However, E802 allele frequencies were significantly higher (50–96% at days 158–167) in saliva specimens during this period and did not decline until casirivimab-imdevimab was administered.

An explanation for these discordant findings is unclear and may relate to the differential compartmentalization of variants within the respiratory tract or the stochastic nature of sampling the nasopharyngeal compartment as opposed to saliva where virus from a wider geographic distribution may pool. A sufficient number of nasopharyngeal and saliva specimens were not obtained to perform a compartmentalization analysis. Given that saliva is infrequently sampled, RDV resistance mediated by E802 mutations may be under-detected. Furthermore, analysis of the 131 genomic sequences that have been found in GISAID to contain the E802D mutation identified two instances where sequences were geographically and temporally clustered, suggesting that local transmission of E802D variants may have occurred (Supplementary Fig. 3). Continued surveillance will thus be needed to elaborate whether the occurrence of E802D, as well as other substitutions at this residue, will be limited because of an underlying fitness cost or alternatively, emerge to pose a broader risk for RDV resistance.

**Additional mutations that were temporally associated with remdesivir therapy.** We did not identify nsp12 mutations at sites other than residue 802 which were temporally associated with RDV therapy. A mutation at D484Y has been reported in a post-treatment sample from an immunocompromised patient who failed RDV therapy[12], but the RDV resistance phenotype associated with this mutation, whose location is distal to the active site, has not been confirmed. We identified two additional mutations, A504V in nsp14 (exonuclease) and I115L in nsp15 (endoRNAse), whose allele frequencies increased during and after RDV treatment (Supplementary Fig. 4). However, A504V in nsp14 was present (allele frequency 85.1%) in a specimen obtained during her initial phase of illness (day 36). I115L in nsp15 was not detected in the early-phase specimen; we therefore cannot exclude the possibility that this mutation in the endoRNAse gene may contribute to RDV resistance. In addition, further investigation is needed to evaluate whether epistatic interactions influence the expression of a remdesivir resistance phenotype beyond the isolated contribution of substitutions at E802D in nsp12.

## Discussion

The case highlights several issues for the management of SARS-CoV-2 infection in immunocompromised patients. First, the course of infection was notable for being indolent despite a six-month period of sustained high-grade viral shedding and underlying lung involvement. We identified six mutations, including a frameshift mutation in ORF3a (N257fs), which were present in the consensus genomes throughout the course of illness prior to RDV treatment (Supplementary Fig. 5). Yet none provides a clear explanation for an attenuated phenotype. Longitudinal profiling of PBMC demonstrated profound depletion of circulating B-cell populations and a T-cell phenotype consistent with chronic antigen exposure as evidenced by the increased activation (CD38+, HLA-DR+) and exhaustion (PD-1, Tim-3) markers (Supplementary Fig. 6, gating strategy in Supplementary Fig. 8). Given reports that maladaptive immune responses contribute to the risk of severe COVID-19[13,14], we speculate that the compromised immune response was unable to control viral load yet conversely, protected the patient from developing severe complications.

Second, SARS-CoV-2 infection was associated with the development of severe refractory neutropenia and moderate anemia in

the patient (Fig. 1c). While we are unable to discern if the cytopenia was due to direct or indirect effects of infection, SARS-CoV-2 has been detected in the bone marrow of an immunocompromised patient who developed pancytopenia during infection[15]. Our patient's cytopenia resolved after casirivimab-imdevimab therapy, suggesting a causal link between SARS-CoV-2 infection and cytopenia in this setting.

Third, the case adds to the evidence from prior studies[4,5,12] that monotherapy with RDV may be associated with a risk for clinical or virologic relapse in immunocompromised patients. Of note, the patient experienced a marked clinical response to RDV therapy, and we were unable to assess whether residual complications (i.e., anosmia, anemia, neutropenia) would have resolved without the intervention of mAb therapy. Nevertheless, the patient developed virologic failure during and after RDV therapy, which prevented her from terminating her prolonged isolation.

Finally, the case illustrates the potential benefit of mAb therapy for immunocompromised patients with persistent SARS-CoV-2 infection. Prior to initiation of mAb therapy, two non-synonymous mutations, H655Y (Supplementary Fig. 7a) and P812L (Supplementary Fig. 7b), emerged which were not present in the initial specimen (day 36), indicating that spike protein continues to evolve and adapt to the host in the absence of selective pressure from an adaptive immune response. One week after casirivimab-imdevimab was administered (day 170), we identified a new spike protein mutation, A348S in the RBD domain, which persisted to day 177 (Supplementary Fig. 7c). WGS was not performed on specimens collected after day 177 because of low amounts or undetectable viral RNA. A348S is not proximal to the mAb binding sites (Supplementary Fig. 7d)[16] and was not identified in a comprehensive analysis of resistance mutations[6]. In the setting of a negligible cellular immune response, mAb therapy cleared viral shedding, abrogated the residual complaint of anosmia, and resolved the blood dyscrasias.

In summary, we identified the de novo emergence of a RDV-resistance mutation, E802D, following initiation of RDV in an immunocompromised patient with persistent SARS-CoV-2 infection. While the finding is limited to a single case and requires confirmation of its generalizability in larger patient populations, it suggests that RDV may impart a selective pressure in vivo to drive evolution of the virus. E802D is associated with a fitness cost in vitro, which may limit the broader impact of this mutation on the development of secondary resistance during treatment and the risk for primary resistance through transmission of resistant variants. Yet, our findings underscore the importance of immunocompromised hosts with uncontrolled viral replication as a source of genetic diversification[3,4,17–19] and selection of mutations that may potentially impart adverse consequences for antiviral therapy. Enhanced genomic surveillance of immunocompromised patients may thus be warranted. As observed in this case, initiation of anti-SARS-CoV-2 mAb may serve as a therapeutic option to achieve rapid and sustained virologic responses and improved clinical outcomes in immunocompromised patients.

## Methods

**Ethics declaration.** Regulatory approval to administer an 8 g dose of casirivimab/imdevimab was obtained from the U.S. Food and Drug Administration under investigational new drug application no. 153390. This study was also approved by Yale Human Research Protection Program Institutional Review Boards (FWA00002571, protocol ID 2000027690, 2000029277). Informed consent was obtained from both the patient and healthcare workers in accordance with CARE guidelines and in compliance with the Declaration of Helsinki principles.

**Isolation of patient plasma and PBMCs.** Patient whole blood was collected in sodium heparin-coated vacutainers and gently agitating at room temperature until sample pick-up by IMPACT team members. All blood was processed on the day of

collection. Plasma samples were collected after centrifugation of whole blood at 400 $g$ for 10 min at room temperature (RT) without brake. The undiluted serum was then transferred to 15-ml polypropylene conical tubes, and aliquoted and stored at −80 °C for subsequent analysis. PBMCs were isolated using Histopaque (Sigma–Aldrich, #10771-500 ML) density gradient centrifugation in a biosafety level 2+ facility. After isolation of undiluted serum, blood was diluted 1:1 in room temperature PBS, layered over Histopaque in a SepMate tube (StemCell Technologies; #85460) and centrifuged for 10 min at 1200 $g$. The PBMC layer was isolated according to the manufacturer's instructions. Cells were washed twice with PBS before counting. Pelleted cells were briefly treated with an ACK lysis buffer for 2 min and then counted. Percentage viability was estimated using standard Trypan blue staining and an automated cell counter (Thermo-Fisher, #AMQAX1000).

**SARS-CoV-2 specific-antibody measurements**. Triton X-100 and RNase A were added to serum samples at final concentrations of 0.5% and 0.5 mg/ml, respectively, and incubated at room temperature (RT) for 30 min as described[20]. 96-well MaxiSorp plates (Thermo Scientific #442404) were coated with 50 μl/well of recombinant SARS Cov-2 S1 protein (ACROBiosystems #S1N-C52H3-100ug) at a concentration of 2 μg/ml in PBS and were incubated overnight at 4 °C. The coating buffer was removed, and plates were incubated for 1 h at RT with 200 μl of blocking solution (PBS with 0.1% Tween-20, 3% milk powder). Serum was diluted 1:50 in dilution solution (PBS with 0.1% Tween-20, 1% milk powder) and 100 μl of diluted serum was added for two hours at RT. Plates were washed three times with PBS-T (PBS with 0.1% Tween-20) and 50 μl of HRP anti-Human IgG Antibody (GenScript #A00166, 1:5,000) diluted in dilution solution added to each well. After 1 h of incubation at RT, plates were washed three times with PBS-T. Plates were developed with 100 μl of TMB Substrate Reagent Set (BD Biosciences #555214) and the reaction was stopped after 15 min by the addition of 2 N sulfuric acid. Plates were read at a wavelength of 450 nm and 570 nm.

**Cell lines and virus**. Vero E6 kidney epithelial cells were cultured in Dulbecco's Modified Eagle Medium (DMEM) supplemented with 1% sodium pyruvate (NEAA) and 5% fetal bovine serum (FBS) at 37 °C and 5% CO2. The cell line was obtained from the ATCC and has been tested negative for contamination with mycoplasma. SARS-CoV-2, strain USA-WA1/2020, was obtained from BEI Resources (#NR-52281) and was amplified in Vero E6 cells. As described[21], cells were infected at a MOI 0.01 for four three days to generate a working stock and after incubation the supernatant was clarified by centrifugation (450 $g$ × 5 min) and filtered through a 0.45-micron filter. The pelleted virus was then resuspended in PBS then aliquoted for storage at −80 °C. Viral titers were measured by standard plaque assay using Vero E6 cells. All experiments were performed in a biosafety level 3 with the Yale Environmental Health and Safety office approval.

**Virus titration from nasopharyngeal swabs**. SARS-CoV-2 titers from longitudinal nasopharyngeal swabs were assessed by conventional plaque assay. Nasopharyngeal swabs were collected and swabs stored and frozen in universal viral transport media (VTM) prior to plaquing. Each nasopharyngeal sample was ten-fold serially diluted (dilution range from 1:10 to 1:1000000). Diluted NP samples were added on a monolayer of Vero E6 cells. After 1 h of infection, DMEM (2% FBS), and 1.2% of Avicel (Dupont, 9004346) were overlaid. 72 h post inoculation, media was removed and 4% Paraformaldehyde added for 1 h. Cells were stained with 0.1% of crystal violet to visualize SARS-CoV-2 plaques. Plaques were counted by single operator. Each sample was tested in duplicate.

**RNA extraction and PCR**. RNA was extracted on a Qiagen Bio Robot (Qiagen, Hilden, Germany) using the QIAamp Virus Bio Robot MDx Kit (Qiagen) following manufacturer's instructions. RT-PCR was performed using AgPath-ID One-Step RT-PCR kit (Life Technologies, Carlsbad, CA, USA) with CDC primers and probes for N1/N2 following protocol and concentrations defined previously[22]. Each 25 μL of reaction mix included 2X RT-PCR buffer, 25X enzyme mix, primers-probes, and 5 μL of the extracted RNA. Cycle parameters on the ABI 7500 real-time PCR system (Applied Biosystems) were 10 min at 48 °C for reverse transcription, 10 min of inactivation at 95 °C, 40 cycles of 15 sec at 95 °C, and 45 sec at 60 °C. Internal control (EXO), positive, and negative (water) control were included on every run.

**Library preparation and sequencing**. Using 11 μL of extracted RNA, single strand complementary DNA (sscDNA) was synthesized using SuperScript IV First-Strand Synthesis System (Thermofisher). Sequencing libraries were prepared using the Swift Normalase Amplicon Panel (SNAP) for SARS-CoV-2 (Swift Biosciences, Ann Arbor, MI, USA) as described previously[8]. Libraries were quantified using fluorometric methods (Quant-iT dsDNA high sensitivity kit, Life Technologies, Carlsbad, CA, USA), normalized, and sequenced on the Illumina NextSeq 500 or 2000 instruments (Illumina, San Diego, CA, USA) using 2 × 150 reads.

**Analysis of sequencing data**. Reads were processed using a custom bioinformatic pipeline (https://github.com/greninger-lab/covid_swift_pipeline), as described[8]. Briefly, raw FASTQs were trimmed of adapter content and low-quality reads, then aligned to the NCBI reference Wuhan-1 (NC_045512). FASTQs are aligned using

BBMap. PCR primers were soft-clipped with Primerclip (https://github.com/swiftbiosciences/primerclip) and a consensus genome was generated from this alignment using bcftools v1.9[23]. All bam files are manually reviewed to ensure accuracy of alignment parameters, because of the potential for unusually large insertions/deletions that can occur in SARS-CoV-2/coronaviruses. Variants were examined longitudinally with a modified Python script adapted from LAVA (https://github.com/greninger-lab/lava).

A k-mer based approach was used to identify reads corresponding to subgenomic RNAs. After removing adapters and low-quality regions, junction-spanning reads corresponding to individual sgRNAs were filtered using BBDuk (https://sourceforge.net/projects/bbmap/) against a custom FASTA file composed of 30 nucleotides of the 5′ leader sequence and 30 nucleotides of the respective 3′ junctions of downstream ORFs[24], allowing an edit distance of 1. In order to avoid calling of genomic reads, we constrained matches to those that had 10 consecutive matches of 31-mers.

Consensus Orf1a/1b sequences were subject to PAML (version 4.8) positive selection analysis as described[9].

**Site-directed mutagenesis and full length icSARS-CoV-2 mNG E802D and E802A production**. The reverse genetics system used to generate E802D and E802A SARS-CoV-2 mutants is described at length elsewhere[11]. The icSARS-CoV-2 mNG (mNeonGreen) backbone previously reported was employed to construct both full-length E802D and E802A mutant viruses utilizing the following primer pairs: forward (E802D): gttggactgaTactgaccttactaaaggac, reverse (E802D): aggtcagtAtcagtccaacattttgcttc; and forward (E802A): gttggactgCgactgaccttactaaaggac, reverse (E802A): aggtcagtcGcagtccaacattttgcttc. Site-directed mutagenesis was performed using Pfu Turbo Polymerase (Agilent) according to manufacturer's instructions. E802D and E802A mutations were confirmed in icSARS-CoV-2 mNG backbone by Sanger sequencing using the primer: gatgatactctctgacgatg.

Steps to produce full-length infectious virus were performed as described previously[11], with the following minor modifications: Plasmid propagation and digestion were performed as described and excision of DNA bands from the gel was done by using a Blue-Light transilluminator (Accuris). Next, equal molar amounts of the seven plasmids were reduced by half relative to the prior reports to perform the full-length genomic ligation: F1 (0.305 μg), F2 (0.325 μg), F3 (0.375 μg), F4 (0.47 μg) were ligated using T4 DNA ligase with a final volume of 50 μl. Plasmids F5 (0.375 μg), F6 (0.36 μg) and F7 (0.3 μg) were incubated with T4 DNase Ligase in a volume of 40 μl. The primary reactions were incubated at 4 °C for 16 h. After this, the two reactions were mixed and volumed up to 100 μl by adding the corresponding additional volume of ligation reagents. The secondary ligation was incubated for 18 h at 4 °C. The full-length genomes were purified by using the Genomic DNA Clean and Concentrator kit from Zymo Research according to manufacturer's instructions. Successful ligation products were visualized using a 0.8% agarose gel and standard gel imager.

Next, full-length in vitro RNA transcription was also performed with additional modifications. First, 500 ng of each ligation product was employed. Reactions were performed using the mMESSAGE mMACHINE T7 Transcription Kit (Thermo Scientific) with a cap analog-to-GTP ratio of 1:1. Incubation of the reaction was done at 32 °C for 12 h. Following DNAse treatment was performed at 37 °C for 15 min. Next, viral RNA was purified by using the RNA Clean and Concentrator kit from Zymo Research according to the manufacturer's instructions. Successful transcription products were visualized using a 0.8% agarose gel and standard gel imager.

RNA transcripts were next electroporated into BHK-N expressing cells (courtesy of Dr. Brett Lindenbach). 15 μg of each mutant transcript was electroporated into BHK cells using the Electro Square Porator Device (BTX). Electroporated cells were seeded into a T75 flask containing 1:2 ratio of BHK-N:Vero-E6 cells. Complete monolayer CPE was evident at 72 h post-electroporation and supernatant was centrifuged and 0.22 um filtered. Vero-E6 were infected with 500 uL of each of the clarified supernatants. Supernatants were collected for each mutant virus at 48 h post infection and tittered as described previously. Infectious clones are available with MTA and appropriate biosafety approvals.

**Sequencing and variant calling**. RNA was extracted from SARS-CoV-2 mutants using the MagMAX viral/pathogen nucleic acid isolation kit and tested with the multiplexed RT-qPCR variant assay[25]. Using a standard dilution series, Ct values were converted to RNA copies per μL. Extracted RNA was diluted to 1 million copies per μL and used as input into the COVIDSeq Test RUO version for library preparation, in duplicate. Pooled libraries were sequenced on the NovaSeq (paired-end 150) at the Yale Center for Genome Analysis. Single nucleotide variants were called at a minimum frequency threshold of 0.02 and minimum read depth of 400X using iVar (version 1.3.1)[26], and filtered between duplicate replicates. Introduction of the E802D and E802A was confirmed at a minimum frequency of 0.82-0.9 in initial stocks.

**Replication kinetics of icSARS-CoV-2 mutants**. Isolates from initial stocks were expanded and used to infect Vero-E6 cells with 0.01 MOI of each E802D and E802A mutant virus to assess their replication kinetics. Supernatants were collected

at 8−, 24−, and 48-hours post-infection. Titers for each time point were assessed by conventional plaque assay as described above. Each virus was assessed in triplicate.

**Remdesivir resistance assay**. Each icSARS-CoV-2 mutant virus (E802D and E802A) was cultured in the presence of increasing concentrations of RDV, ranging from 0.39 to 40 μM. Infections were performed using Vero-E6 in 384-well plates containing the corresponding RDV dilutions in phenol-red free DMEM media with 5% FBS and incubated at 37 °C for 4 h. Next, cells were infected with 0.01 MOI of each mutant virus and infected cell frequencies were measured at 48 h post-infection by mNeonGreen expression by high content imaging (Cytation 5, BioTek)[27]. Cell culture supernatants were also collected for plaque assay. Cell viability in uninfected cells was assessed at 72 h post-infection using the CellTiter-Glo kit (Promega) according to the manufacturer's instructions.

**Phylogenetic analysis**. Phylogenic trees were generated using the Nextstrain software package (version 3.0.6)[28]. 326 background sequences from CT, USA collected April 1st-September 30th 2020 were downloaded from GISAID [www.gisaid.org] (Supplementary Table 3). The tree was rooted using the Wuhan reference strain. Genome sequences which contain the E802D mutation were identified from the GISAID database (accessed on 10/31/21; Supplementary Table 4) and were compared to sequences in the global Auspice dataset.

**Flow cytometry**. Flow cytometry was performed as described[21]. Antibody clones and vendors used for flow cytometry analysis were as follows: BB515 anti-hHLA-DR (G46-6) (1:400) (BD Biosciences), BV605 anti-hCD3 (UCHT1) (1:300) (Bio-Legend), BV785 anti-hCD4 (SK3) (1:200) (BioLegend), APCFire750 or PE-Cy7 or BV711 anti-hCD8 (SK1) (1:200) (BioLegend), BV421 anti-hCCR7 (G043H7) (1:50) (BioLegend), AlexaFluor 700 anti-hCD45RA (HI100) (1:200) (BD Biosciences), PE anti-hPD1 (EH12.2H7) (1:200) (BioLegend), APC anti-hTIM3 (F38-2E2) (1:50) (BioLegend), BV711 anti-hCD38 (HIT2) (1:200) (BioLegend), BB700 anti-hCXCR5 (RF8B2) (1:50) (BD Biosciences), PE-Cy7 anti-hCD127 (HIL-7R-M21) (1:50) (BioLegend) and PE-CF594 anti-hCD25 (BC96) (1:200) (BD Biosciences). In brief, freshly isolated PBMCs were plated at $1–2 × 10^6$ cells per well in a 96-well U-bottom plate. Cells were resuspended in Live/Dead Fixable Aqua (ThermoFisher) for 20 min at 4 °C. Following a wash, cells were blocked with Human TruStan FcX (BioLegend) for 10 min at RT. Cocktails of desired staining antibodies were added directly to this mixture for 30 min at RT. For reinfection stains, cells were first washed and supernatant aspirated; then to each cell pellet a cocktail of secondary markers was added for 30 min at 4 °C. Prior to analysis, cells were washed and resuspended in 100 μl 4% PFA for 30 min at 4 °C. Following this incubation, cells were washed and prepared for analysis on an Attune NXT (ThermoFisher). Data were analysed using FlowJo software version 10.6 software (Tree Star). The specific sets of markers used to identify each subset of cells are summarized in Supplementary Figs. 6 and 8.

**Reporting summary**. Further information on research design is available in the Nature Research Reporting Summary linked to this article.

## Data availability

Sequencing data are available under NCBI BioProject no. PRJNA774781. GenBank accession numbers for consensus genomes are provided in Supplementary Table 2. GISAID [www.gisaid.org] sequences incorporated into phylogenetic analyses are identified in Supplementary Tables 3–4. Source data are provided with this paper.

## Code availability

The custom bioinformatic pipeline is available at https://doi.org/10.5281/zenodo.6142073 (https://doi.org/10.5281/zenodo.6142073)[29].

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

## Acknowledgements

The authors thank Dr. Michael Jacobs, Dr. Sanjay Bhagani and Dr. David Lowe at the Royal Free Hospital for their advice regarding the management of the patient, Dr. Jennifer Hamilton, David Stein, Dr. Wendy Kampman and Dr. David M. Weinreich at Regeneron Pharmaceuticals, Inc. for their guidance on the use of monoclonal antibody therapy and laboratory evaluations, Nathanial Price at Yale School of Medicine for his assistance in identifying data and samples for the analyses, and Dr. Chantal Vogels, Dr. Joseph Fauver, Mallery Breban and Dr. Nathan Grubaugh at the Yale School of Public Health for sequencing infectious clones and providing advice on the analysis and interpretation of the genomic data. Casirivimab and imdevimab was provided by Regeneron Pharmaceuticals, Inc. for the purpose of emergency compassionate use treatment. We also thank the health care staff at Yale New Haven Hospital, which include Dr. Jensa Morris, Dr. Anne Spichler, Dr. Frederick Altice and Dr. Nikhil Seval, who cared for the patient during her hospital course and provided key input, and most of all, the patient for her equanimity and contributions to this study. This study was supported by the Beatrice Kleinberg Neuwirth Fund; the Sendas Family Fund, Yale

School of Public Health; and Department of Internal Medicine at the Yale School of Medicine. SG was supported by a grant from the National Institutes of Health (2T32AI007517-21A1). C.B.W. was supported by Burroughs Welcome Fund, Ludwig Family Foundation, and Mathers Foundation.

## Author contributions

J.K., A.J.R. and M.P.H. contributed to data collection, analysis, and writing. M.L., P.R., J.F., D.F., C.V., M.C.M., S.M.B., E.W., N.K., P.L., W.W., A.P. contributed to data collection and analysis. B.L. provided reagents and contributed to study design. O.O., A.G., C.B.W., A.I. and W.L.S. conceived of the study and contributed to study design, interpretation, and editing of the manuscript. S.G. and A.I.K. conceived of the study and contributed to study design, data interpretation, writing, and regulatory approvals.

## Competing interests

A.L.G. reports institutional central testing contracts from Abbott and research grants from Merck and Gilead, outside of the proposed work. A.I.K. received consulting fees from Tata Sons and is the recipient of grants on COVID-19 from Merck, Regeneron and Serimmune, all of which are outside the submitted work. O.O. received consulting fees from Gilead and ViiV, as well as research support and honoraria from Gilead, outside of the submitted work. W.L.S. is a consultant for Hugo Health, founder of Refactor Health and is recipient of grants on COVID-19 from Merck and Regeneron, all of which are outside the submitted work. The remaining authors declare no competing interests.
