## [Peer Review File · Nature Communications]

nature portfolio

Peer Review File

Draft OnlyREVIEWER COMMENTS

Reviewer #1 (Remarks to the Author):

The manuscript “De novo emergence of a remdesivir resistance mutation during treatment of persistent SARS-CoV-2 infection in an immunocompromised patient”, by Gandhi et al, is a case report of a patient with non-Hodgkin’s lymphoma with prolonged COVID-19 symptoms and high-grade SARS-CoV-2 viral shedding, with demonstrated development of an nsp12 mutation while on RDV therapy that has previously been shown in vitro to confer RDV resistance. The authors demonstrate that the emergence of this mutation while on therapy coincided with rebound of SARS-CoV-2 viral load, percent subgenomic RNA, and PFU, and they then show that infection of Vero cells in the laboratory with a clone engineered to contain the mutation results in in vitro RDV resistance, as well as decreased viral fitness in the absence of RDV. They also perform an analysis of the frequency and distribution of the mutation in the global GISAID database.

The paper is well-written and is novel in that it provides the first in vivo example of development of an RDV resistance mutation while on therapy. Its strengths include its phylogenetic analysis showing that all strains detected originated from the patient’s initial viral strain; its longitudinal performance of WGS, with sufficient frequency to demonstrate convincingly that the mutation of interest developed while on therapy; and its in vitro demonstration that the mutation identified does in fact confer RDV resistance. The finding that viral fitness is decreased in vitro is interesting and provides a plausible explanation for why this mutation has not become more widespread; the GISAID analysis provides evidence that strains with acquired RDV resistance can be transmitted and thus argues for the importance of the finding. The clinical case presented is of interest in its own right as an example of high-grade viral shedding with persistent clinical symptoms, without evidence of severe illness and in a patient who had not received immunosuppressive therapy in over a year at the time of presentation. Minor comments are provided below.

Specific comments:

1. It appears from figure 1 that some parameters, including fever, anosmia, CRP, and CT chest findings, did have sustained improvement after RDV therapy, despite the emergence of the RDV-resistant strain and rebound of Ct values to pre-RDV levels. It may be worth commenting on this specifically and providing some proposed explanations (simply a lag between viral rebound and clinical symptoms? Or is it possible that the in vitro less-fit virus is also less clinically virulent?)
2. Figure 1d: it might be helpful if this was also presented horizontally so that all parameters over time are presented similarly.
3. The GISAID analysis, which is an interesting addition to the paper, suggests that remdesivir resistance by this mechanism is a very rare phenomenon. It could be useful to comment in the

discussion on whether there have been any clinical reports of initial response to remdesivir with rebound of either symptoms or Ct values while still on treatment.

4. The patient received a 10 day course of remdesivir. Can the authors comment on why the patient received 10 days instead of 5 days? Do the authors think this more prolonged course played a role in selection of RDV resistance? I am not aware of any resistance data from ACTT-1 or from the SIMPLE trial, but it would be helpful for the authors to explicitly comment on this.

Reviewer #2 (Remarks to the Author):

The authors present findings on de novo Remdesivir resistance mutations arising in an immunocompromised individual with NHL. This was a well-written and engaging case study and of general interest due to the widespread use of Remdesivir prior to new therapies being widely available. The use of REGNRON in this patient is also highly topical. I commend the authors on their findings, though have several comments to make the manuscript stronger:

Major Comments:

1) Multiple tissue samples were collected (Nasopharyngeal, saliva, stool and blood) and the authors refer to potential compartmentalisation. Phylogenies indicate some separation between stool, saliva and nasopharyngeal samples – I would like to see some formal compartmentalisation analysis conducted to see if populations were vastly different as new evidence suggests that there are major differences between sampling sites and the authors suggest ongoing intra-host diversification.

2) E802D is a mutations known in the literature and authors suggest that RDV may provide a selection pressure for this mutation. Can you perform a formal selection analysis to determine if indeed any of the sites you notice are under a selection pressure? An example would be to try PAML (Ziheng Yang) and corroborate your findings.

3) Replication kinetics experiments utilise only Vero E6 cells. Literature has shown marked differences in SARS-CoV-2 kinetics, dependent on cell lines and/or organoids used. I would like to see experiments repeated using human airway epithelial cells or Calu-3 to confirm if the results observed are consistent between cell lines, especially as samples were collected from nasopharyngeal swabs (representing different compartments).

Minor comments:

1) How were the fastqs aligned? Some literature make use of mafft, using the --keeplength parameter which removes insertions/deletions relative to the Wuhan strain which could confound mutations.

2) Figure S8 shows the gating strategy for flow cytometry, though I can see now reference to this in the manuscript?

Reviewer #3 (Remarks to the Author):

This is a well written manuscript about a carefully performed case-study that provides good arguments to postulate that the E802D mutation in the SARS-CoV2 RNA dependent polymerase is associated with remdesivir therapy failure. The study is important and of relevance to the field.

I have only a minor comment which is about the Legend of Fig S2c; there the following is stated "Significance of infected cells with SARS-CoV-2 WT (top panel) or E802D/A (bottom panel)"; we assume that this should be phrased as (for example) : "Viability of non-infected cells (top panel) or cells infected with SARS-CoV-2 wild type or E802D/A (bottom panel)E802D/A (bottom panel)"

Reviewer #4 (Remarks to the Author):

This case report describes the clinical course of an immunocompromised COVID-19 patient with persistent SARS-CoV-2 infection who received remdesivir (RDV) treatment followed by a monoclonal antibody cocktail. The viral titer briefly declined after initiation of RDV treatment, but then sharply increased halfway through treatment. To determine if the genetic makeup of the virus suggested RDV resistance, virus-containing specimen were sequenced over time, and mutations identified. A E802D mutation was identified in nsp12, the molecular target of RDV, along with multiple other mutations in nsp12 and elsewhere in the genome. Of these, three replicase mutations were temporally associated with the increase in viral loads. The nsp12 E802D mutation was studied by itself using recombinant virus. E802D conferred resistance to RDV in cultured cells as well as reduced replicative fitness in the absence of RDV in cultured cells. This is the first report that describes the emergence of nsp12 and other mutations in a clinical specimen from a RDV -treated immunocompromised patient. The work emphasizes the importance of monitoring for remdesivir

resistance and the potential benefit of combinatorial therapies in immunocompromised patients with SARS-CoV-2 infection.

To adequately interpret the effect of variations in genetic makeup of an isolate on its sensitivity to RDV (and any other direct-acting antiviral), they need to be studied in the context of the genetic makeup of the virus that evolved under the circumstances. The nsp12 E802D mutation arose in combination with other nonsynonymous mutations located within other replicase genes and elsewhere in the genome, not by itself. According to the list shown in FigS4, a large number of nonsynonymous mutations emerged during the course of treatment. Each one of these can influence viral fitness and sensitivity to RDV. This information is lost by studying the E802D mutation in isolation, as the authors have done here. We know from the work by Szemiel et al. (PMCID: PMC8496873) that compensatory mutations can influence the sensitivity to the drug: the seemingly unrelated nsp6 mutation in combination with nsp12 E802D actually decreased the resistance of SARS-CoV-2 to RDV compared to the nsp12 E802D mutation alone. Therefore, the authors' statement "Although I115L in nsp15 was not detected in the early-phase specimen, the plausibility of a mutation in the endoRNase gene which confers RDV resistance is less clear." is not supported by the literature. In addition, the presence of the nsp14 mutation at high frequency early on during infection does not preclude it from having an effect. Some mutations only arise in combination with or following the appearance of other mutations that enhance viral fitness. This could explain why the frequency of the mutation in the viral population increased despite its apparent fitness cost. I recommend the authors study the mutations in combination before suggesting a causal relationship between the emergence of one nsp12 mutation and the increase in viral loads during RDV treatment, as the virus population that emerged during treatment may not reflect the resistance phenotype observed for the single mutant in vitro.

It is not clear why the experiments were performed in Vero E6 cells instead of well-established lung cell lines such as Calu3 or A549-hACE2 expressing cells. Besides lacking physiological relevance, Vero E6 cells are known to poorly metabolize RDV to its active triphosphate form (see PMCID: PMC7340027). Perhaps the choice of cell type contributes to the discrepancy in the magnitude of the resistance phenotypes reported by Szemiel et al. and the current study. Another factor could be the use of a reporter as a readout for viral replication instead of infectious viral titer. Use of a reporter as a readout for viral replication may overestimate the effect of the mutation on RDV sensitivity compared to infectious viral titer or genome copy number readouts. The effect of the E802A mutation on RDV sensitivity observed in the mNG reporter assays does not appear to be reflected in a marked change in infectious viral particle production (Fig S2b) or cell viability (Fig S2c).

Fig S4: It would be helpful to know what the frequencies of these mutations are to understand which ones increase in frequency during treatment and which ones do not, as was done for the spike mutations in figure S7. Also, nsp12 E802D, nsp14 A540V, and nsp15 I115L are not represented in the table.

The sentence “These findings suggest that RDV resistance mediated by substitutions at residue 802 in nsp12 may lead to therapeutic failure in the setting of an immunocompromised host.” suggest a causal relationship between the single mutation studied in this work and therapeutic failure. The data presented in the current version of the manuscript do not support causation.

Draft Only

Responses to Reviewers Comments

Reviewer #1 (Remarks to the Author):

The manuscript “De novo emergence of a remdesivir resistance mutation during treatment of persistent SARS-CoV-2 infection in an immunocompromised patient”, by Gandhi et al, is a case report of a patient with non-Hodgkin’s lymphoma with prolonged COVID-19 symptoms and high-grade SARS-CoV-2 viral shedding, with demonstrated development of an nsp12 mutation while on RDV therapy that has previously been shown in vitro to confer RDV resistance. The authors demonstrate that the emergence of this mutation while on therapy coincided with rebound of SARS-CoV-2 viral load, percent subgenomic RNA, and PFU, and they then show that infection of Vero cells in the laboratory with a clone engineered to contain the mutation results in in vitro RDV resistance, as well as decreased viral fitness in the absence of RDV. They also perform an analysis of the frequency and distribution of the mutation in the global GISAID database.

The paper is well-written and is novel in that it provides the first in vivo example of development of an RDV resistance mutation while on therapy. Its strengths include its phylogenetic analysis showing that all strains detected originated from the patient’s initial viral strain; its longitudinal performance of WGS, with sufficient frequency to demonstrate convincingly that the mutation of interest developed while on therapy; and its in vitro demonstration that the mutation identified does in fact confer RDV resistance. The finding that viral fitness is decreased in vitro is interesting and provides a plausible explanation for why this mutation has not become more widespread; the GISAID analysis provides evidence that strains with acquired RDV resistance can be transmitted and thus argues for the importance of the finding. The clinical case presented is of interest in its own right as an example of high-grade viral shedding with persistent clinical symptoms, without evidence of severe illness and in a patient who had not received immunosuppressive therapy in over a year at the time of presentation. Minor comments are provided below.

We thank the reviewer for their comments regarding the novelty and strengths of our manuscript. We address their specific points below.

Specific comments:

1. It appears from figure 1 that some parameters, including fever, anosmia, CRP, and CT chest findings, did have sustained improvement after RDV therapy, despite the emergence of the RDV-resistant strain and rebound of Ct values to pre-RDV levels. It may be worth commenting on this specifically and providing some proposed explanations (simply a lag between viral rebound and clinical symptoms? Or is it possible that the in vitro less-fit virus is also less clinically virulent?)

We thank the reviewer for this insight and modified the text to note that the patient did have a clinical response to RDV (lines 82-85).

2. Figure 1d: it might be helpful if this was also presented horizontally so that all parameters over time are presented similarly.

We agree with this comment and have adjusted the layout for Figure 1d accordingly.

3. The GISAID analysis, which is an interesting addition to the paper, suggests that remdesivir resistance by this mechanism is a very rare phenomenon. It could be useful to comment in the discussion on whether there have been any clinical reports of initial response to remdesivir with rebound of either symptoms or Ct values while still on treatment.

We are unaware of patients who have had clinical or virological relapse while on treatment. We updated our discussion to cite reports (references 4, 5, 11) of clinical and virologic relapse that has occurred in immunosuppressed patients after completion of therapy (lines 199-204)

4. The patient received a 10 day course of remdesivir. Can the authors comment on why the patient received 10 days instead of 5 days? Do the authors think this more prolonged course played a role in selection of RDV resistance? I am not aware of any resistance data from ACTT-1 or from the SIMPLE trial, but it would be helpful for the authors to explicitly comment on this.

The duration of remdesivir therapy was extended given concerns for clinical and virologic relapse after remdesivir. We added this clarification (lines 80-82). It is unlikely that the extended course contributed to selection of resistance since the E802D mutants were identified on Day 7 after initiation of remdesivir.

Reviewer #2 (Remarks to the Author):

The authors present findings on de novo Remdesivir resistance mutations arising in an immunocompromised individual with NHL. This was a well-written and engaging case study and of general interest due to the widespread use of Remdesivir prior to new therapies being widely available. The use of REGNRON in this patient is also highly topical. I commend the authors on their findings, though have several comments to make the manuscript stronger:

We thank the reviewer for their supportive comments.

Major Comments:

1) Multiple tissue samples were collected (Nasopharyngeal, saliva, stool and blood) and the authors refer to potential compartmentalisation. Phylogenies indicate some separation between stool, saliva and nasopharyngeal samples – I would have like to see some formal compartmentalisation analysis conducted to see if populations were vastly different as new evidence suggests that there are major differences between sampling sites and the authors suggest ongoing intra-host diversification.

We unfortunately did not have a sufficient number of specimens in each compartment to perform a compartmentalization analysis with sufficient statistical power. We placed this caveat in the text (lines 160-161).

2) E802D is a mutations known in the literature and authors suggest that RDV may provide a selection pressure for this mutation. Can you perform a formal selection analysis to determine if

indeed any of the sites you notice are under a selection pressure? An example would be to try PAML (Ziheng Yang) and corroborate your findings.

We performed PAML analysis of the entire Orf1a/1ab open reading frame and found that nsp12 E802 had the highest probability (BEB Prob($w>1$), 0.984; mean omega, 9.90) for positive selection among residues that were newly identified after remdesivir administration.

3) Replication kinetics experiments utilise only Vero E6 cells. Literature has shown marked differences in SARS-CoV-2 kinetics, dependent on cell lines and/or organoids used. I would like to see experiments repeated using human airway epithelial cells or Calu-3 to confirm if the results observed are consistent between cell lines, especially as samples were collected from nasopharyngeal swabs (representing different compartments).

We chose to perform our replication kinetics in Vero E6 cells as they are widely used to assay kinetics for SARS-CoV-2 and other human coronaviruses¹. Additionally, we utilized these cells to establish whether there was an intrinsic replication defect in Nsp12 E802D mutants (explaining its relatively low prevalence among SARS-CoV-2 genomes) that might be otherwise masked in alternative cell lines. The use of our control WA01-mNG virus without mutations ensures that we can detect intrinsic replication defects conferred by both E802D and E802A mutations.

1. Wang, M., Cao, R., Zhang, L., Yang, X., Liu, J., Xu, M., Shi, Z., Hu, Z., Zhong, W., Xiao, G. 2020. Remdesivir and chloroquine effectively inhibit the recently emerged novel coronavirus (2019-nCoV) in vitro. *Cell Research*. 30: 269-271.

Minor comments:

1) How were the fastqs aligned? Some literature make use of mafft, using the --keeplength parameter which removes insertions/deletions relative to the Wuhan strain which could confound mutations.

FASTQs were aligned using BBMap. All bam files are manually reviewed to ensure accuracy of alignment parameters, because of the potential for unusually large insertions/deletions that can occur in SARS-CoV-2/coronaviruses. We included the above on lines 264-266 in our methods

2) Figure S8 shows the gating strategy for flow cytometry, though I can see now reference to this in the manuscript?

We added text reference to Figure S8 in the manuscript (line 190)

Reviewer #3 (Remarks to the Author):

This is a well written manuscript about a carefully performed case-study that provides good arguments to postulate that the E802D mutation in the SARS-CoV2 RNA dependent polymerase is associated with remdesivir therapy failure. The study is important and of relevance to the field.

I have only a minor comment which is about the Legend of Fig S2c; there the following is stated

“Significance of infected cells with SARS-CoV-2 WT (top panel) or E802D/A (bottom panel)”; we assume that this should be phrased as (for example) : “Viability of non-infected cells (top panel) or cells infected with SARS-CoV-2 wild type or E802D/A (bottom panel)E802D/A (bottom panel)”

We thank the reviewer for their supportive review and summary of our work. We clarified the legend text for Figure S2c.

Reviewer #4 (Remarks to the Author):

This case report describes the clinical course of an immunocompromised COVID-19 patient with persistent SARS-CoV-2 infection who received remdesivir (RDV) treatment followed by a monoclonal antibody cocktail. The viral titer briefly declined after initiation of RDV treatment, but then sharply increased halfway through treatment. To determine if the genetic makeup of the virus suggested RDV resistance, virus-containing specimen were sequenced over time, and mutations identified. A E802D mutation was identified in nsp12, the molecular target of RDV, along with multiple other mutations in nsp12 and elsewhere in the genome. Of these, three replicase mutations were temporally associated with the increase in viral loads. The nsp12 E802D mutation was studied by itself using recombinant virus. E802D conferred resistance to RDV in cultured cells as well as reduced replicative fitness in the absence of RDV in cultured cells. This is the first report that describes the emergence of nsp12 and other mutations in a clinical specimen from a RDV -treated immunocompromised patient. The work emphasizes the importance of monitoring for remdesivir resistance and the potential benefit of combinatorial therapies in immunocompromised patients with SARS-CoV-2 infection.

We thank the reviewer for their thorough summary of our work and address specific comments below.

To adequately interpret the effect of variations in genetic makeup of an isolate on its sensitivity to RDV (and any other direct-acting antiviral), they need to be studied in the context of the genetic makeup of the virus that evolved under the circumstances. The nsp12 E802D mutation arose in combination with other nonsynonymous mutations located within other replicase genes and elsewhere in the genome, not by itself. According to the list shown in FigS4, a large number of nonsynonymous mutations emerged during the course of treatment. Each one of these can influence viral fitness and sensitivity to RDV. This information is lost by studying the E802D mutation in isolation, as the authors have done here. We know from the work by Szemiel et al. (PMCID: PMC8496873) that compensatory mutations can influence the sensitivity to the drug: the seemingly unrelated nsp6 mutation in combination with nsp12 E802D actually decreased the resistance of SARS-CoV-2 to RDV compared to the nsp12 E802D mutation alone. Therefore, the authors' statement “Although I115L in nsp15 was not detected in the early-phase specimen, the plausibility of a mutation in the endoRNase gene which confers RDV resistance is less clear.” is not supported by the literature.

We agree with the reviewer's comments that epistatic interactions among mutations may influence susceptibility to RDV and revised the text (lines 176-178) to state the possibility.

Of note, the reviewer may be referring to Figure S5 and not Figure S4. Figure S4 shows the prevalence of only two mutations, A504V in nsp14 and I115L in nsp15, which changed while on RDV therapy. Figure S5 shows mutations that emerged prior to initiation of therapy. We updated the title and legend of Figure S5 to make this distinction clearer.

In addition, the presence of the nsp14 mutation at high frequency early on during infection does not preclude it from having an effect. Some mutations only arise in combination with or following the appearance of other mutations that enhance viral fitness. This could explain why the frequency of the mutation in the viral population increased despite its apparent fitness cost. I recommend the authors study the mutations in combination before suggesting a causal relationship between the emergence of one nsp12 mutation and the increase in viral loads during RDV treatment, as the virus population that emerged during treatment may not reflect the resistance phenotype observed for the single mutant in vitro.

We agree with this reviewer that mutations can be both conditional and synergistic in nature and as in the response above, cited the possibility of epistatic interactions (lines 178-180).

It is not clear why the experiments were performed in Vero E6 cells instead of well-established lung cell lines such as Calu3 or A549-hACE2 expressing cells. Besides lacking physiologically relevance, Vero E6 cells are known to poorly metabolize RDV to its active triphosphate form (see PMID: PMC7340027). Perhaps the choice of cell type contributes to the discrepancy in the magnitude of the resistance phenotypes reported by Szemiel et al. and the current study. Another factor could be the use of a reporter as a readout for viral replication instead of infectious viral titer. Use of a reporter as a readout for viral replication may overestimate the effect of the mutation on RDV sensitivity compared to infectious viral titer or genome copy number readouts. The effect of the E802A mutation on RDV sensitivity observed in the mNG reporter assays does not appear to be reflected in a marked change in infectious viral particle production (Fig S2b) or cell viability (Fig S2c).

Please see response to Reviewer #2 (critique 3). As RDV also effectively inhibits our control virus under our experimental conditions using Vero E6 cells, and as Vero E6 cells have been extensively used in other studies investigating RDV, we believe that sufficient functional RDV is present to interpret our data. While we agree that the mNG reporter virus measures translation and not replication, it has been widely used as a surrogate of viral burden in previous studies.¹ We also note that we directly measured infectious viral titers (Figure S2b), confirming our results regarding E802D-conferred RDV resistance.

1. Xie, X., Muruato, A., Lokugamage, K., Narayanan K., Zhang, X., Zou, J., Liu, J., Schindewolf, C., Bopp, N., Aguilar, P., Plante, K., Weaver, S., Makino, S., LeDuc, J., Menachery, V., Shi, Pei-Yong. 2020. An Infectious cDNA clone of SARS-CoV-2. *Cell Host & Microbe*. 27: 841-848.

Fig S4: It would be helpful to know what the frequencies of these mutations are to understand which ones increase in frequency during treatment and which ones do not, as was done for the spike mutations in figure S7. Also, nsp12 E802D, nsp14 A540V, and nsp15 I115L are not

represented in the table.

As noted above, we believe that the reviewer is referring to Figure S5, which is a table of mutations in the consensus genomes (ie allele frequency >50%) that were present before initiation of remdesivir treatment. We have updated the table title and legend to clarify that these mutations were identified prior to treatment. We also updated the text (line 185) to clarify that these are the consensus genomes. As the nsp12 E802D and nsp15 I115L mutations were detected after treatment, they were not included in this table but rather were included Figure S4 where their allele frequencies are graphed.

The sentence “These findings suggest that RDV resistance mediated by substitutions at residue 802 in nsp12 may lead to therapeutic failure in the setting of an immunocompromised host.” suggest a causal relationship between the single mutation studied in this work and therapeutic failure. The data presented in the current version of the manuscript do not support causation.

Our data establishes a clear temporal link between remdesivir administration and mutant emergence in an immune compromised host. We obtained multiple lines of evidence that support the plausibility of a causal relationship. We feel that the body of evidence presented in the manuscript supports the assertion that these finding “suggest” a causal relationship.”

Draft Only